# Identification and Pharmacological Characterization of a Low-Liability Antinociceptive Bifunctional MOR/DOR Cyclic Peptide

**DOI:** 10.3390/molecules28227548

**Published:** 2023-11-11

**Authors:** Yangmei Li, Shainnel O. Eans, Michelle Ganno-Sherwood, Abbe Eliasof, Richard A. Houghten, Jay P. McLaughlin

**Affiliations:** 1College of Pharmacy, University of South Carolina, Columbia, SC 29208, USA; aeliasof@email.sc.edu; 2Department of Pharmacodynamics, College of Pharmacy, University of Florida, Gainesville, FL 32610, USA; shaieans@cop.ufl.edu; 3Torrey Pines Institute for Molecular Studies, Port St. Lucie, FL 34987, USA; condor1082@aol.com (M.G.-S.); houghten@tpims.org (R.A.H.)

**Keywords:** mixed-based combinatorial library, structure–activity relationships, cyclic peptide, mu-opioid receptor, delta-opioid receptor, bifunctional ligand, antinociception, opioid antagonist, opioid liabilities

## Abstract

Peptide-based opioid ligands are important candidates for the development of novel, safer, and more effective analgesics to treat pain. To develop peptide-based safer analgesics, we synthesized a mixture-based cyclic pentapeptide library containing a total of 24,624 pentapeptides and screened the mixture-based library samples using a 55 °C warm water tail-withdrawal assay. Using this phenotypic screening approach, we deconvoluted the mixture-based samples to identify a novel cyclic peptide Tyr-[D-Lys-Dap(Ant)-Thr-Gly] (CycloAnt), which produced dose- and time-dependent antinociception with an ED_50_ (and 95% confidence interval) of 0.70 (0.52–0.97) mg/kg *i.p*. mediated by the mu-opioid receptor (MOR). Additionally, higher doses (≥3 mg/kg, *i.p.*) of CycloAnt antagonized delta-opioid receptors (DOR) for at least 3 h. Pharmacological characterization of CycloAnt showed the cyclic peptide did not reduce breathing rate in mice at doses up to 15 times the analgesic ED_50_ value, and produced dramatically less hyperlocomotion than the MOR agonist, morphine. While chronic administration of CycloAnt resulted in antinociceptive tolerance, it was without opioid-induced hyperalgesia and with significantly reduced signs of naloxone-precipitated withdrawal, which suggested reduced physical dependence compared to morphine. Collectively, the results suggest this dual MOR/DOR multifunctional ligand is an excellent lead for the development of peptide-based safer analgesics.

## 1. Introduction

The perception of pain arises from a variety of nociceptive insults, each transduced and transmitted through distinct systems to which specific analgesics may provide varying degrees of relief [1]. Opioid analgesics have a long history of use to relieve moderate to extreme pain [1]. However, repeated opioid use results in clinical liabilities such as the development of tolerance and dependence and a potentially lethal respiratory depression [2]. The misuse of opioid analgesics leads to the increasing risk of opioid abuse and overdose, which have contributed to the escalating economic burden imposed on society. Thus, there is a critical public health need for novel, safer, and more efficacious analgesics to treat pain. 

One strategy to address this need focuses on the development of opioid peptides into safer analgesics [3,4,5]. The opioid system utilizes opioid receptors and the endogenous opioid peptides to modulate pain, yet the endogenous opioid system is not observed to produce respiratory depression and addictive side effects. A recent study using a genetically encoded biosensor revealed the mu-opioid peptide receptor (MOR) and the delta-opioid peptide receptor (DOR) not only function at the cell membrane, but also inside the cell at different locations, depending on the chemistry of ligand types [6]. After peptides activated the MOR in the plasma membrane, MOR quickly internalized and was detected in an active state in endosome for 20 min, while opiate alkaloids penetrated the membrane to produce Golgi-localized receptor activation after MOR was activated in the plasma membrane [6]. The spatiotemporal specificity is thought to affect signal duration and pathway selection downstream, contributing to distinct downstream physiological effects [7,8] As the endogenous opioid peptides, but not opiate alkaloids, can precisely modulate pain, the different spatiotemporal patterns of opioid receptor activation may link to the severity of adverse effects limiting opiate use in pain management [9]. 

The endogenous opioid peptides such as enkephalins, endorphins, and dynorphins exert their analgesic effects through the MOR, DOR, and kappa (KOR) opioid receptors. These peptides have varying affinities at the opioid receptors and none of them bind exclusively to one opioid receptor type [10]. For example, enkephalins bind and activate the DOR and MOR. Endorphins exert their effect primarily through the MOR, but they also have affinity for the DOR. While dynorphins have high affinity for the KOR, they have significant affinity for the MOR and DOR as well. Most importantly, as a whole, the endogenous opioid peptides activate all opioid receptors.

All three opioid receptors mediate opioid-dependent analgesia. While MOR has been the primary target for clinically used opioid analgesics, activation of MOR may lead to respiratory depression and physical dependence. Unlike MOR, activation of DOR does not cause respiratory depression and physical dependence [11]. Since DOR is mainly located in the cytoplasm, with a very low level of receptors being associated with the plasma membrane [12], activation of DOR alone has been reported to produce insufficient analgesia in a model of acute pain [11]. However, mounting evidence indicates that the activation of DOR produces analgesia in various models of chronic inflammatory and neuropathic pain [13,14]. 

Importantly, *multifunctional* ligands, which simultaneously modulate MOR and DOR, have been demonstrated to produce enhanced antinociception with fewer adverse effects [15,16,17,18,19,20]. For example, co-administration of the DOR agonist SNC80 with the MOR agonist heroin enhanced the antinociceptive effect of heroin while decreasing the sedative and reinforcing effects [21]. Dual agonists effective at both MOR and DOR, such as MMP-2200 [22,23] and SRI-22141 [24], produce high antinociceptive potency as well as decreasing the propensity to produce locomotor stimulation, tolerance, and physical dependence in mouse models of chronic pain. Simultaneous activation of MOR and DOR using a dual agonist LP2 not only significantly ameliorated mechanical allodynia, but also produced neuroprotective effects in a rodent model of neuropathic pain [25]. Interestingly, dual-function ligands acting as a MOR agonist/DOR antagonist have also been reported to produce enhanced antinociceptive efficacy with reduced tolerance and dependence [15,26,27]. DIPP-NH_2_[Ψ] [28], SoRi20411 [29], and 14-alkoxy pyridomorphinans [30] are potent DOR antagonists and MOR agonists which produce analgesia with reduced tolerance when administered repeatedly at the ED_80_ antinociceptive dose supraspinally twice daily for 5−7 days. Ligands simultaneously modulating multiple opioid receptors, such as MOR and DOR, therefore hold great promise for the development of safe analgesics.

We have previously reported the synthesis of a mixture-based cyclic peptide library consisting of 24,624 anthraniloyl(Ant)-tagged pentapeptides in a positional scanning format (Figure 1) [31]. The fully defined cyclic peptide Tyr-[D-Lys-Dap(Ant)-Tyr-Gly] identified from this library demonstrated high affinity for the MOR in a competition receptor-binding assay [31]. As cyclic peptides have improved stability against proteolysis, we hypothesized a cyclic peptide with a desired antinociceptive, safety, and pharmacokinetic profile would be found within the library. To identify such a cyclic peptide analgesic from this library, we employed a direct phenotypic *in vivo* screening using the 55 °C warm water tail-withdrawal (WWTW) assay to facilitate the discovery of the analgesic lead [32,33]. As the WWTW assay measures the overall antinociception of a sample in mice, the antinociceptive lead compounds discovered should have a favored pharmacokinetic profile without a preconceived bias for the mechanism of action [34]. Using this direct *in vivo* screening, we identified a fully defined, novel, potent, low-liability cyclic peptide analgesic from the mixture library. Further target identification characterization using opioid receptor knock-out mice elucidated the analgesic lead produced agonist activity at MOR, while pharmacological testing demonstrated DOR antagonism, indicating that it acted as a MOR agonist/DOR antagonist in mice.

## 2. Results

### 2.1. Mixture-Based Cyclic Peptide Library

As we reported previously, the cyclic peptides in the library [31] were synthesized from peptide thioesters [35] using an imidazole-promoted cyclization strategy [36]. All the pentapeptides in this library have a Gly fixed at R^1^ position and a Dap(Ant) at fixed R^3^ position. The rest of the R^2^, R^4^, and R^5^ positions are made either with a single amino acid (*O*) or a mixture of amino acids (*X*) [31]. The library was composed of the R^2^-, R^4^-, and R^5^-sub-libraries (Figure 1). R^2^-position and R^4^-position were constituted by a total of 36 amino acids (AAs) including 19 *L*- and 17 *D*-amino acids, while R^5^-position was generated with 19 natural *L*-amino acids. Thus, there are 36 R^2^ samples in the R^2^-sublibrary, 36 R^4^ samples in the R^4^-sublibrary, and 19 R^5^ samples in the R^5^-sublibrary; each sample is a mixture of pentapeptides. Each sample contains a total of 684 (36 × 19) pentapeptides at the R^2^- and R^4^-sublibraries, and 1296 (36 × 36) pentapeptides at the R^5^-sublibrary. This positional scanning library has a total of 91 (36 + 36 + 19) samples and 24,624 (36 × 36 × 19) pentapeptides in the linear sequences.

### 2.2. Library Deconvolution Using Direct In Vivo Screening

Utilizing a positioning scanning format, the library was screened *in vivo* via antinociception assessed with the 55 °C WWTW assay, as we previously described [32,33]. Each partially defined, mixture-based sample was administered intraperitonially (*i.p.)* at a dose of 5 mg/kg. Additional mice received vehicle alone (10%DMSO/90% saline, *i.p*.) as a negative control, or morphine (10 mg/kg, *i.p*.) as a positive control. The combined time mice demonstrated for withdrawment of their tails for each sample tested was summed over six time points, and was reported by substitution position (Figure 2A–C). Analysis of the responses with one-way ANOVA detected significant global responses in the R^2^-(F_(37,266)_ = 12.1, *p* < 0.0001; Figure 2A), R^4^-(F_(37,266)_ = 13.1, *p* < 0.0001; Figure 2B) and R^5^-sublibraries (F_(25,182)_ = 9.50, *p* < 0.001; Figure 2C). Post hoc analysis (Dunnett’s) identified samples from the R^2^-, R^4^- and R^5^-sublibraries producing a combined average response time to tail-withdrawal that was both significantly different from the vehicle response (*p* < 0.05 vs. dotted line, Figure 2A–C), but not different from the morphine response (*p* > 0.05 vs. cyan bar, Figure 2A–C). Meeting these conditions for the R^2^- position were samples 5 (Gly), 6 (His), 11 (Asn), 12 (Pro), 13 (Gln), 16 (Thr) and 25 (D-Ile), whereas for the R^4^ position, only Sample 62 (D-Lys) showed this response, while the R^5^ position demonstrated three samples meeting the criteria: 75 (Glu), 81 (Leu) and 91 (Tyr).

To validate the antinociceptive effects of the five strongest partially defined samples and to determine a potential dose–response relationship, additional doses of 10 mg/kg or 25 mg/kg, *i.p.* of the selected samples were administered to mice and tested *in vivo* using the 55 °C warm water tail-withdrawal assay (Figure 3). Samples 5, 6 and 16 defined at the R^2^-position produced dose-dependent antinociception that peaked at 60 min post administration (Figure 3A). Estimated from the limited doses tested, antinociceptive ED_50_ (and 95% confidence intervals) values were near that of morphine (4.12 (2.73–5.82) mg/kg, *i.p.*), at 13.0 (10.2–17.8) mg/kg, *i.p.* (Sample 5), 17.5 (12.5–33.2) mg/kg, *i.p.* (Sample 6) and 7.15 (3.91–9.96) mg/kg, *i.p.* (Sample 16). Similarly, mixture-based samples 81 and 91 defined at the R^5^-position produced antinociceptive ED_50_ (and 95% confidence intervals) values of 3.62 (0.00–8.58) mg/kg, *i.p.* (Sample 81) and 7.15 (0.52–12.7) mg/kg, *i.p.* (Sample 91) at their peak effect of 120 min post administration (Figure 3B). While limited to estimates, this data collectively confirms the antinociceptive effects of the partially defined, mixture-based samples, validating further deconvolution of the library. Taken together, Thr (16) at R^2^, D-Lys (62) at R^4^, and Tyr (91) and Leu (81) at R^5^ were selected as the most active residues at each position to deconvolute the mixture-based library for the identification of the cyclic peptide that produced the most significant antinociceptive activity in mice with the 55 °C WWTW assay.

### 2.3. Synthesis of Cyclic Peptides Tyr-[D-Lys-Dap(Ant)-Thr-Gly] (CycloAnt) and Leu-[D-Lys-Dap(Ant)-Thr-Gly] (CycloAnt-Leu) for the Deconvolution of the Library

Two cyclic peptides, Tyr-[D-Lys-Dap(Ant)-Thr-Gly] (CycloAnt) and Leu-[D-Lys-Dap(Ant)-Thr-Gly] (CycloAnt-Leu), were then synthesized to complete the deconvolution of the library. Using mercaptopropyl isobutyl polyhedral oligomeric silsesquioxane (POSS-SH) as a soluble support [37], Boc-protected amino acids including Gly (R^1^), Thr(tBu) (R^2^), Dap[Ant(Boc)] (R^3^), D-Lys(Alloc) (R^4^), and Tyr(tBu)/Leu (R^5^) were subsequentially coupled to form the peptide POSS-thioesters, Boc-Tyr(tBu)-D-Lys(Alloc)-Dap(Ant)-Thr-Gly-S-POSS and Boc-Leu-D-Lys(Alloc)-Dap(Ant)-Thr-Gly-S-POSS, respectively (Figure 1). The peptide POSS-thioesters were then treated with Pd(PPh_3_)_4_ in the presence of PhSiH_3_ in DCM to remove the Alloc protective group, followed by adding imidazole to carry out a one-pot, in situ cyclization to form the final side-to-tail cyclic peptides, CycloAnt and CycloAnt-Leu, respectively [38].

The cyclic peptides were then treated with 55% TFA to remove the protective groups and the resulting compounds were purified by RP-HPLC and structurally characterized by NMR experiments before characterization of their pharmacological profile. 

### 2.4. Screening of the Cyclic Peptide CycloAnt for Protein Interaction

The potential interaction of CycloAnt with a panel of 47 protein targets, including GPCRs, kinases, nuclear hormone receptors, ion channels, and enzymes, was assessed by assays included in the SAFETYscan service from Eurofins DiscoverX. CycloAnt was tested in both agonist and antagonist modes for each representative GPCRs (see Appendix A). Tested in a cAMP assay utilizing cells overexpressing each receptor, CycloAnt demonstrated MOR and DOR agonism, with EC_50_ values of 14.0 and 3.36 nM, respectively (Appendix A). CycloAnt did not produce agonist or antagonist activity at any other target tested in this panel.

### 2.5. Evaluation of the Cyclic Peptides for Their Antinociceptive Effects in Wild-Type Mice

We tested the antinociceptive effects of the cyclic peptides using the 55 °C WWTW assay with C57BL/6J male mice (Figure 4). Both peptides produced dose- (Figure 4A) and time-dependent antinociception (F_(16,152)_ = 38.5; *p* < 0.0001; two-way ANOVA; Figure 4B). Peak antinociception was produced by both peptides within 20 min of administration, with a significant duration of 70 min for CycloAnt (3 mg/kg, *i.p.*) and 50 min for the CycloAnt-Leu analog (100 mg/kg, *i.p.*) compared to vehicle (*p* < 0.05, Dunnett’s post hoc test; Figure 4B). The two cyclic peptides produced full antinociception, but with significant differences in potency. CycloAnt-Leu produced antinociception with an ED_50_ (and 95% C.I.) value of 38.3 (18.7–127.1) mg/kg, *i.p.*, 9.3-fold less potent than the control opioid morphine (ED_50_ (and 95% C.I.) value = 4.12 (2.73–5.82) mg/kg, *i.p.*; Figure 4A). In contrast, CycloAnt displayed an ED_50_ (and 95% C.I.) value of 0.70 (0.52–0.97) mg/kg, *i.p.*, 5.9-fold more potent than morphine. Collectively, this data demonstrates CycloAnt is a potent antinociceptive “hit” identified through the direct phenotypic *in vivo* screening and deconvolution of this mixture-based cyclic peptide library.

### 2.6. Evaluation of Opioid Receptor Involvement in CycloAnt Antinociception

To assess the contributions of the individual opioid receptors to CycloAnt-mediated antinociception, the cyclic peptide (3 mg/kg, *i.p.*) was administered to mice lacking the mu- (MOR KO), kappa- (KOR KO) or delta-opioid receptor (DOR KO). Compared to the response in wild-type C57BL/6J mice, CycloAnt-mediated antinociception was significantly reduced 80% only in MOR KO mice (F_(3,28)_ = 216.3, *p* < 0.0001, one-way ANOVA with Dunnett’s post hoc test), and not KOR KO or DOR KO mice (Figure 5).

### 2.7. Evaluation of Opioid-Receptor-Selective Antagonism Mediated by CycloAnt

Given the MOR and DOR activity identified in the *in vitro* screening, and the singular MOR agonist activity displayed in mice, CycloAnt was evaluated for antagonist activity against the KOR-selective agonist U50,488 (10 mg/kg, *i.p.*), and the DOR-selective agonist SNC-80 (100 nmol, *i.c.v.*; Figure 6A). MOR KO mice were used to preclude complications of the mu-opioid receptor-mediated antinociception. U50,488 antinociception was not significantly antagonized by pretreatment with CycloAnt (3 or 10 mg/kg, *i.p.*; F_(2,21)_ = 0.9, *p* = 0.42, one-way ANOVA; Figure 6A left bars). However, surprisingly, CycloAnt pretreatment significantly antagonized SNC-80 antinociception in a dose-dependent manner (F_(2,21)_ = 22.3, *p* < 0.0001, one-way ANOVA with Dunnett’s post hoc testing; Figure 6A, right bars). Examined at the dose of maximal DOR antagonism (10 mg/kg, *i.p.*), CycloAnt-mediated DOR antagonism was time dependent (F_(4,43)_ = 16.2, *p* < 0.0001, one-way ANOVA with Dunnett’s post hoc testing; Figure 6B), with significant reduction of SNC-80 antinociception for 1 or 3 h (*p* < 0.0001), but not 6 h (*p* = 0.07) or 8 h (*p* = 0.96). This data suggests CycloAnt acts *in vivo* as a multifunctional MOR agonist and DOR antagonist.

### 2.8. Evaluation of CycloAnt for Respiratory and Hyperlocomotor Effects

Respiratory depression is a major life-threatening effect of clinically used opioid analgesics. To evaluate whether the cyclic peptide-based MOR/DOR multifunctional ligand can dissociate this side effect from antinociception, we examined CycloAnt for its potential to depress breathing rate over a 120 min period using the automated, computer-controlled Comprehensive Lab Animal Monitoring System (CLAMS) apparatus in male C57BL/6J mice [34]. As expected, the positive control morphine (10 or 30 mg/kg, *i.p.*) produced significant dose and time-dependent respiratory depression compared to vehicle F_(20,360)_ = 6.67, *p* < 0.001, two-way RM ANOVA with Tukey’s multiple comparison post hoc test; Figure 7A). In contrast, CycloAnt did not significantly impair breathing when administrated *i.p.* at either the peak antinociceptive dose (3.0 mg/kg) or a higher dose 15 times the antinociceptive ED_50_ value (10.6 mg/kg), and in fact statistically increased it in a time and dose-dependent manner up to 100 min compared to vehicle (*p* < 0.05, Tukey’s test; Figure 7A). 

As a MOR agonist, morphine induces psychostimulation demonstrated by hyperlocomotion in mice. Consistent with this, treatment significantly increased ambulation (factor: time × treatment, F_(20,360)_ = 36.7, *p* < 0.0001, two-way RM ANOVA; Figure 7B), primarily due to morphine (factor: treatment, F_(10,250)_ = 34.8, *p* < 0.001, two-way RM ANOVA). In contrast, CycloAnt induced significant hyperlocomotor activity (factor: treatment, F_(10,205)_ = 3.19, *p* = 0.0008, two-way RM ANOVA with Tukey’s test; Figure 7B), but CycloAnt-induced ambulations were significantly less than those induced by morphine after 20 min to the end of the testing period († *p* < 0.05, Tukey’s test; Figure 7B).

### 2.9. Evaluation of CycloAnt for Potential of Opioid-Induced Hyperalgesia (OIH), Tolerance and Dependence Development

Chronic use of opioid analgesics may cause opioid-induced hyperalgesia, the paradoxical response where a patient becomes more sensitive to painful stimuli, as well as antinociceptive tolerance and physical dependence. To assess this, mice were treated twice daily for four days with the morphine (10 mg/kg, *i.p.*) or CycloAnt (3 mg/kg, *i.p*.), and once more on the 5th day before WWTW testing. Tail-withdrawal latencies were collected both prior to the start of treatment (on day 1), and again on the morning of day 5 prior to drug treatment. Each group of mice showed equivalent initial responses on day 1 to a 48 °C warm water stimulus (morphine, 10 ± 1.14 s vs. CycloAnt, 9.1 ± 0.68 s; *p* = 0.52, Student’s *t*-test; Figure 8A) or 55 °C warm water stimulus (morphine, 1.43 ± 0.05 s vs. CycloAnt, 1.56 ± 0.08 s; *p* = 0.20, Student’s *t*-test), whereas chronic morphine administration induced significant hyperalgesia (F_(1,17)_ = 14.3, *p* = 0.002; two-way RM ANOVA) against a 48 °C warm water stimulus on day 5 (with a tail withdrawal latency of 4.35 ± 0.47 s; *p* < 0.0001, Sidak’s post hoc test; Figure 8A), chronic CycloAnt treatment did not induce significant OIH (day 5 tail withdrawal latency of 9.02 ± 0.83 s; *p* = 0.997, Sidak’s post hoc test; Figure 8A). However, repeated chronic administration of either morphine or CycloAnt over 4 days produced significant reductions in the tail-withdrawal latency of the compound on day 5 (F_(1,18)_ = 5.75, *p* = 0.03; two-way REML ANOVA with Sidak’s post hoc test; Figure 8B), indicating opioid-induced tolerance.

We then examined the potential development of physical dependence after chronic administration of morphine or CycloAnt, as above, followed by precipitation of withdrawal with naloxone. Repeated treatment with escalating doses of morphine induced physical dependence, demonstrated by administration of naloxone (Table 1). Morphine produced significant increases in the frequency of forepaw tremor, presence of diarrhea, and frequency of jumping and teeth chattering, along with decreases in rearing and the forepaw licking frequency characteristic of opioid physical dependence (see one-way ANOVA values in Table 1). In contrast, while CycloAnt showed significant increases in diarrhea and reduced rearing frequency compared to saline control mice, most other signs of withdrawal, such as jumping frequency and teeth chattering, significantly differed from those of morphine or were completely absent (Table 1). Collectively, these results indicate chronic CycloAnt treatment induced much less physical dependence than that seen with morphine.

## 3. Discussion

From the direct *in vivo* phenotypic screening of the mixture-based cyclic peptide library using the 55 °C WWTW assay, we have identified a novel hit CycloAnt, Tyr-[D-Lys-Dap(Ant)-Thr-Gly], exhibiting time- and dose-dependent antinociception with an ED50 value of 0.7 mg/kg (*i.p.*) in mice. Notably, this library was screened earlier using a receptor-binding assay for the MOR, identifying a hit with a binding affinity of 14 nM [31]. However, that cyclic peptide has a sequence of Tyr-[D-Lys-Dap(Ant)-Tyr-Gly], differing from the sequence Tyr-[D-Lys-Dap(Ant)-Thr-Gly] identified in this study. The differing results may stem from the two screening methods utilized, based on different mechanisms and readouts. First, as the *in vivo* direct phenotypic screening tests checked the relative antinociceptive effect of a sample, in terms of the MOR, only agonists can be identified from the assay. While the binding assay at MOR measures the affinity of a sample at the receptor, whether the sample is an agonist or antagonist cannot be differentiated, leaving activity unresolved. Second, the present samples were administered to mice through the intraperitoneal route for the *in vivo* direct phenotypic screening. As the mice are complete organisms with systemic physiology and functions such as metabolism, only these samples with a preferred pharmacokinetic profile will succeed in this screening. It seems likely that the Tyr-[D-Lys-Dap(Ant)-Tyr-Gly] previously identified using the receptor binding assay is unable to adequately activate the MOR under physiological conditions in a complete organism, precluding antinociception in the mouse WWTW assay.

The mixture-based cyclic peptide library is constituted using the imidazole-catalyzed cyclization of linear pentapeptide. While the goal is to cyclize all the linear peptides in a head-to-tail pattern, the composition of the mixture-based positional library is more complicated, as side-to-tail cyclization can take place when Lys or D-Lys is in the linear sequence. Lys and D-Lys are used at positions 2 and 4, and Lys is used at position 5 in the linear samples; therefore, the peptides in the library are not only cyclized in the head-to-tail pattern, but also has the side-to-tail as a side product when Lys or D-Lys is in the sequence. Since we are tracking the antinociceptive effect and not the purity of the compound in the library, as long as the side reaction is reproducible, the active compounds can be identified through the deconvolution of the mixture library. In this case, CycloAnt, a side-to-tail cyclic peptide, which produced a potent antinociceptive effect, has been successfully identified through the phenotypic screening in mice. 

CycloAnt interactions with a broad set of molecular targets were determined with the Safety47 Panel Dose Response, using the SAFETYscan provided by Eurofins DiscoverX. This panel utilizes 78 assays at 47 protein targets, including GPCRs, kinases, nuclear hormone receptors, ion channels, and enzymes. Notably, a number of targets associated with antinociception are included in this panel, including MOR, KOR, and DOR, cyclooxygenase 1 (COX1), cyclooxygenase 2 (COX 2), human serotine receptors 1A (5-HT1A), 1B (5-HT1B), 2A (5-HT2A), and 2B (5-HT2B), human adrenoceptors 3A (5-HT3A), alpha2A (ADRA1), beta1 (ADRB1), and beta2 (ADRB2), voltage gated sodium channel (Nav1.5), and the human glutamate ion channel (NMDAR). CycloAnt only interacted with MOR and DOR in the panel, showing agonist activity at both receptors using the cAMP assay. It did not produce agonist or antagonist activity at any other pain-associated targets in this panel. While the results of MOR testing were confirmed with our *in vivo* testing presently, they diverged with the surprising demonstration *in vivo* of DOR antagonism. Notably, the DOR agonist activity in the cAMP assay was made with cells expressing a high level of DOR, whereas the *in vivo* results utilizing mice with a physiologically normal level of DOR expression indicated the compound behaved as an antagonist. This discrepancy on the function at DOR may be accounted for receptor reserve [40,41,42], as a very weak partial agonist/antagonist can appear as a full agonist when receptor reserve is present. While the discrepancy is beyond the scope of this initial identification and characterization, future investigations may prove beneficial in better understanding this interaction. Regardless, consistent with the SAFETYscan results, CycloAnt was not found to produce any significant biological activity at any other targets in this study than DOR and MOR. 

The combination of DOR antagonism with MOR agonism has been reported to produce antinociception while possessing safety advantages over typical MOR agonists alone [43]. Although a reduction in antinociceptive tolerance has been shown via a co-administration of MOR agonists with DOR antagonists [44,45] or via a bifunctional MOR agonist/DOR antagonist compounds [46], chronic administration of CycloAnt presently produced a reduction of subsequent antinociception, which was consistent with significant tolerance. However, chronic exposure to CycloAnt did not produce the significant opioid-induced hyperalgesia demonstrated by morphine, and displayed reduced signs of opioid physical dependence, consistent with earlier pharmacological studies [44,45] and reports where *i.c.v*. pretreatment with DOR-selective antisense oligonucleotides decreased morphine dependence in mice [47,48]. It is also consistent with the study in DOR knock out mice in which morphine retained its MOR-mediated analgesic activity without producing tolerance and dependence upon chronic administration [49]. Importantly, CycloAnt did not suppress breathing rate at high doses up to 15 times of the antinociceptive ED_50_ value, and, in fact, increased breathing rate significantly. CycloAnt treatment also induced significantly less hyperlocomotion than that observed with even lower doses of morphine. All of these results are consistent with the recently reported activity of the bifunctional MOR agonist/DOR antagonist [Nal(2′)^4^]CJ-15,208, which also did not produce hyperlocomotion or respiratory depression; indeed, the bifunctional peptide briefly and significantly increased respiration as well [46]. Given the absence of CycloAnt-mediated activity at off-target proteins, future studies are anticipated to investigate potential mechanisms by which the addition of DOR antagonism may contribute to the improved safety advantages demonstrated by this cyclic peptide.

The potent antinociceptive effect and low liability profile after the *i.p*. administration suggests this cyclic peptide has a preferred pharmacodynamic and pharmacokinetic profile for systemic use as a potentially safer pain modulator. Therefore, this MOR agonist/DOR antagonist is an excellent initial lead for the development of a peptide-based safer opioid analgesic.

## 4. Materials and Methods

### 4.1. General Materials

Silica gel, 130–270 mesh, 60 Å, BET surface area 500 m^2^/g, pore volume 0.75 cm^3^/g, was purchased from Sigma-Aldrich Chemical Company Inc., St. Louis, MO, USA. Boc amino acid derivatives, benzotriazole-1-yloxy-trispyrrolidino-phosphonium hexafluorophosphate (PyBOP), N,N-diisopropylethylamine (DIEA), and trifluoroacetic acid (TFA) were purchased from Chem-Impex. HF were purchased from Air Products (San Marcos, CA, USA). Polypropylene mesh (74 micro) was utilized to prepare mesh packets. Phenyltrimethoxysilane and *p*-Chloromethylphenyltrimethoxysilane were purchased from Gelest, Inc. (Tullytown, PA, USA). Mercaptopropyl isobutyl POSS (POSS-SH) was purchased from Hybrid Plastics (Hattiesburg, MS, USA). Pd(PPh_3_)_4_ and all solvents were purchased from Fisher Scientific, Hampton, NH, USA. 

LC-MS (APCI and ESI) was recorded on an Agilent LCMS-1260 (Santa Clara, CA, USA) infinity II at 6120 quadrupole LC/MS mass spectrometer at 214 nm using a Zorbax Eclipse Plus C18 (3.0 × 100 mm, 3.5 μm) column. Preparative RP-HPLC was performed on a Shimadzu preparative HPLC using a Phenomenex Luna 10 μ C18 100 Å column (21.2 × 250 mm). NMR spectra were recorded on a Bruker Avance III-HD 400 instrument (Singapore) at 400 MHz for ^1^H NMR and 100 MHz for ^13^C NMR. NMR chemical shifts are expressed in ppm relative to internal solvent peak and coupling constant were calculated in hertz.

### 4.2. Synthesis of the Mixture-Based Cyclic Peptide Library

The cyclic peptide library was synthesized as we have previously reported [31]. Briefly, all linear peptide thioesters were synthesized using the functionalized mercaptomethylphenyl silica gel as ‘volatilable’ support. Boc-amino acids were coupled on the silica support using PyBOP/DIEA as coupling reagents. *X*-positions were coupled as mixtures of N^α^-Boc protected amino acids using concentration ratios to compensate for the relative reaction rates in competitive couplings. Position 3 was coupled with a N^α^-Boc-N^β^-Alloc-diaminoproponic acid. After peptide elongation on resin, the Boc-protected peptides were treated with Pd(PPh_3_)_4_ (0.1 equiv.) in the presence of PhSiH_3_ (20 equiv.) in DCM to remove the Alloc protection group. The resin bound peptides were coupled with 2-nitrobenzoic acid, followed by the treatment of 2 M SnCl_2_ in DMF overnight to reduce the nitro group to an amino group to generate the Ant-label. After removal of the Boc group with 55% TFA, the resin-bound Ant-labeled peptides were treated with anhydrous HF in the presence of 5% anisole at 0 °C for 2 h. After removal of the HF with nitrogen stream and lyophilization, the linear peptide thioesters were cyclized in a mixed solution of 1.5 M imidazole (aq.) and acetonitrile (1:7 *v*/*v*) at a concentration of 1 mM for 72 h [36], forming the Ant-labeled cyclic peptides.

### 4.3. Solution-Phase Synthesis of Cyclic Peptides Tyr-[D-Lys-Dap(Ant)-Thr-Gly] (CycloAnt) and Leu-[D-Lys-Dap(Ant)-Thr-Gly] (CycloAnt-Leu)

#### 4.3.1. Synthesis of Boc-Dap[Ant(Boc)]-OH

All Boc-amino acids used for the POSS-thioester synthesis were commercially available, except the third amino acid, Boc-Dap[Ant(Boc)], which was synthesized in the lab and directly utilized for the peptide synthesis without further purification. Briefly, Boc-Dap-OH (1 mmol) was dispersed in 2 mL of DMF with the presence of DIEA (1 mmol). In a separate vial, Boc-2-aminobenzoic acid (1 mmol, 0.11 M in DCM) was mixed with PyBOP (1 mmol) and DIEA (1.5 mmol) and activated for 5 min. The solution of Boc-Dap-OH was then added to the activated Boc-2-aminobenzoic acid. The reaction was stirred at r.t. and the completion of the reaction was monitored by LC-MS. After 5 h, the resulting Boc-Dap[Ant(Boc)] was utilized directly for peptide synthesis without purification.

#### 4.3.2. Synthesis of Linear Peptide POSS-Thioesters Boc-Tyr(tBu)-D-Lys(Alloc)-Dap(Ant)-Thr-Gly-S-POSS and Boc-Leu-D-Lys(Alloc)-Dap(Ant)-Thr-Gly-S-POSS

The linear peptide thioester was synthesized using POSS-SH as a soluble support as we previously described [37]. Briefly, Boc-Gly-OH (1.05 mmol) pre-activated with PyBOP (1.05 mmol) and DIEA (2.1 mmol) was added to POSS-SH (0.10 M in 10 mL DCM). The reaction mixture was stirred at room temperature for 3 h. After removal of DCM, Boc-Gly-S-POSS was washed 3 times with 75% acetonitrile containing aqueous HCl (0.05 M). The resulting precipitation was collected by centrifugation and freeze-dried. Boc-Gly-S-POSS was treated with 55% TFA for 30 min to remove the Boc protective group. After removal of TFA/DCM *in vacuo*, the peptide chain was elongated by coupling to the subsequent Boc-amino acid using the method described above. At each of the peptide coupling steps, an extra amount of DIEA was added to the reaction mixture to adjust the pH value to 8. The completion of the amino acid coupling reaction was monitored by a ninhydrin test. The linear peptide POSS-thioesters were obtained at an overall total yield of 81.8%.

#### 4.3.3. Synthesis of Cyclic Peptides Tyr-[D-Lys-Dap(Ant)-Thr-Gly] (CycloAnt) and Leu-[D-Lys-Dap(Ant)-Thr-Gly] (CycloAnt-Leu)

The cyclic peptides were synthesized using the imidazole-promoted cyclization methods as we previously reported [38]. Under N_2_ protection, the Boc-Tyr(tBu)-D-Lys(Alloc)-Dap(Ant)-Thr-Gly-S-POSS (1.0 mmol) or the Boc-Leu-D-Lys(Alloc)-Dap(Ant)-Thr-Gly-S-POSS (1.0 mmol) was dissolved in 100 mL of DCM to a concentration of 10 mM. A portion of PhSiH_3_ (24 mmol) and Pd(PPh_3_)_4_ (0.07 mmol) were then added to facilitate the removal of the Alloc protective group. After reacting overnight, imidazole (1.36 g, 20 mmol) was directly added to the reaction mixture to a concentration of 0.2 M and the reaction was carried out for 24 h. After removal of the solvent *in vacuo*, 75% acetonitrile (aq.) was added to dissolve the yielded cyclic peptide while precipitating the POSS-SH released from peptide cyclization. The supernatant was collected and then lyophilized to dry. The resulting powder was washed with water to remove imidazole. The precipitation was collected and re-dissolved in 65% acetonitrile. After centrifugation at 4000 rpm for 10 min, the supernatant was collected and freeze-dried to yield the protected cyclic peptide Boc-Tyr(tBu)-[D-Lys-Dap(Ant)-Thr-Gly] or Boc-Leu-[D-Lys-Dap(Ant)-Thr-Gly]. Following the treatment with 55% TFA for 30 min, the fully unprotected cyclic peptide Tyr-[D-Lys-Dap(Ant)-Thr-Gly] (CycloAnt) or Leu-[D-Lys-Dap(Ant)-Thr-Gly] (CycloAnt) was yielded. The crude compound was purified by reverse-phase HPLC. The cyclic peptide was obtained as a TFA salt after HPLC purification with a purity of 95%.

Tyr-[D-Lys-Dap(Ant)-Thr-Gly] (CycloAnt). ^1^H NMR (400 MHz, DMSO-*d*_6_) *δ* 0.99–1.22 (m, 2H), 1.03 (d, 3H, *J* = 6 Hz), 1.08–1.18 (m, 1H), 1.29–1.39 (m, 3H), 1.49–1.56 (m, 1H), 2.85–2.87 (m, 1H), 2.90 (m, 1H), 2.96–2.98 (m, 1H), 3.08–3.12 (m, 1H), 3.50 (td, 1H, *J* = 14, 6 Hz), 3.88 (d, 1H, J = 6.8 Hz), 3.90–3.94 (m, 1H), 3.99–4.02 (m, 2H), 4.08 (t, 1H, *J* = 6 Hz), 4.2–4.24 (m, 1H), 5.07 (d, 1H, *J* = 4.8 Hz), 6.34 (br.s, 2H), 6.49–6.52 (dd, 1H, *J* = 7.2, 1 Hz), 6.54 (d, 1H, *J* = 2 Hz), 6,72 (d, 3H, *J* = 8 Hz), 7.03 (d, 2 H, *J* = 8 Hz), 7.16 (td, 1H, *J* = 7.6, 1.6 Hz), 7.33 (t, 1H, *J* = 5.2 Hz), 7.46 (dd, 1H, *J* = 8, 1.2 Hz), 7.64 (d, 1H, *J* = 5.2 Hz), 7.72 (br.s, 1H), 8.13 (br.s, 2H), 8.29 (t, 1H, *J* = 5.6 Hz), 8.38 (t, 1H, *J* = 5.6 Hz), 8.43 (d, 1H, *J* = 7.2 Hz), 8.47 (d, 1H, *J* = 6.4 Hz), 9.39 (s, 1H). ^13^C NMR (100 MHz, DMSO-*d*_6_) *δ* 20.1, 21.2, 28.1, 31.5, 36.8, 38.2, 40.7, 43.8, 53.6, 54.2, 56.4, 60.2, 66.3, 114.9, 115.3, 115.7, 115.8, 116.9, 118.7, 125.2, 128.8, 130.9, 132.5, 149.7, 157.1, 158.4, 158.7, 168.2, 169.2, 170.5(2), 170.6, 172.1. ^19^F NMR (376 MHz) *δ*-37.7 (s). [M + H]^+^ calculated 655.3; found 655.3

Leu-[D-Lys-Dap(Ant)-Thr-Gly] (CycloAnt-Leu). ^1^H NMR (400 MHz, DMSO-*d_6_*) *δ* 0.92 (d, 6H, *J* = 8 Hz), 1.03 (d, 3H, *J* = 4 Hz), 1.29–1.32 (m, 2H), 1.38–1.39 (m, 2H), 1.57–1.65 (m, 5H), 2.96–2.99 (m, 1H), 3.12–3.16 (m, 1H), 3.36 (dd, 1H, *J* = 16, 4 Hz), 3.53 (m, 1H), 3.88 (dd, 1H, *J* = 16, 8 Hz), 3.93–4.00 (m, 1H), 4.02–4.04 (m, 2H), 4.12 (t, 1H, *J* = 8 Hz), 4.19 (t, 1H, *J*= 8 Hz), 6.51 (td, 1H, *J* = 8, 1 Hz), 6.73 (d, 1H, *J* = 8 Hz), 7.16 (td, 1 H, *J* = 8, 1 Hz), 7.41 (t, 1H, *J* = 8 Hz), 7.48 (dd, 1H, *J* = 7.6, 1.6 Hz), 7.66 (d, 1H, *J* = 8 Hz), 8.15 (br.s, 3H), 8.26 (t, 1H, *J* = 6 Hz), 8.38 (t, 1H, *J* = 5.6 Hz), 8.57 (d, 1H, *J* = 8 Hz), 8.83 (d, 1H, *J* = 6.8 Hz). ^13^C NMR (100 MHz, DMSO-*d*_6_) *δ* 20.1, 21.7, 22.1, 23.2, 24.1, 28.2, 31.6, 38.2, 40.4, 40.5, 43.8, 51.4, 54.4, 56.4, 60.0, 66.3, 114.7, 115.2, 115.8, 116.9, 118.8, 128.8, 132.5, 149.8, 158.4, 158.8, 169.2, 169.5, 170.5(2), 170.7, 172.4. [M + H]^+^ calculated 605.3; found 605.3.

### 4.4. Animals

Male C57BL/6J mice (19–27 g each, Jackson Laboratories, Bar Harbor, ME, USA) were used for screening and deconvoluting the library. Male mu-opioid receptor gene knockout mice (MOR KO), kappa-opioid receptor gene-knockout mice (KOR KO), and delta-opioid receptor gene-knockout mice (DOR KO), obtained from a breeding colony established at the University of Florida with progenitors from Jackson Labs, were used for additional testing of the lead. Four mice per cage were housed in a room with controlled temperature and 12 h light/dark cycle controls with the lights on from 07:00 a.m. to 19:00 p.m.; food and water were available ad libitum. All procedures with mice were approved by the local IACUC.

### 4.5. Assessment of Opioid-Induced Hyperalgesia Using the 48 °C Tail-Withdrawal Assay in Mice

The 48 °C warm water tail-withdrawal assay was performed as previously described [39], with the latency to withdraw the tail taken as the end point. The water temperature of 48 °C was selected for this work to ensure a moderate tail-withdrawal response, with a measurable decrease in withdrawal time possible, but also a significant temperature for hyperalgesic testing. If the mouse failed to display a tail-flick in 30 s, the tail was removed from the water to minimize tissue damage. After determining control latencies on day 1, mice received chronic morphine (10 mg/kg, *i.p.*) or CycloAnt (3 mg/kg, *i.p.*) treatment twice a day for 4 days, as described below. Dosing of these compounds was based on the demonstration of equianalgesic efficacy (see Section 2). Mice were then tested prior to treatment on day 5 for evidence of induced hyperalgesia. Experimentally induced decreases in tail-withdrawal latencies in this assay indicate effects of hyperalgesia [50,51].

### 4.6. Antinociceptive Test Using the 55 °C Tail-Withdrawal Assay in Mice

The 55 °C warm water tail-withdrawal assay was performed, as previously described, [52] to screen the library [32]. Mixed-based samples were dissolved at a concentration of 5 mg/mL in saline containing 10% DMSO. Each sample was given to mice (*n* = 8) intraperitoneally (*i.p.*) at a dose of 5 mg/kg. At 0.5, 1, 2, 3.5, 5, 8 h post-administration of the samples, the tail-withdrawal latencies were measured for each sample. The average response time to tail-withdrawal at the six time points were then combined. The combined response time to withdrawal tail was used to determine the antinociceptive effect of each sample. A cutoff of 15 s was used to avoid tissue damage; those mice failing to withdraw their tails within this time were assigned a maximal antinociceptive score of 100%. 

To determine antinociceptive activity of individual compounds, the tail-withdrawal latency was determined repeatedly every 10 min, following administration of the compounds for 1 h or until latencies returned to baseline values. At each time point, antinociception was calculated according to the following formula: % antinociception = 100 × (test latency − control latency)/(15 − control latency). Tail-withdrawal data points are the means of 8–16 mice, unless otherwise indicated, with SEM shown by error bars. 

The opioid receptor involvement in the agonist activity of the individual compounds was determined by testing in MOR KO, KOR KO and DOR KO mice as utilized previously [53].

To determine antagonist activity, MOR KO mice were pretreated with CycloAnt 20 min prior to the administration of the KOR-selective agonist U50,488 (10 mg/kg, *i.p.*), or DOR-selective agonist SNC-80 (100 nmol, *i.c.v.*); the MOR KO mice do not display antinociceptive activity of CycloAnt. Antinociception produced by the established opioid agonists was then measured 40 min after their administration. Additionally, the duration of DOR antagonism produced by CycloAnt was further determined by evaluating SNC-80-induced antinociception 3, 6 or 8 h after administration of CycloAnt.

### 4.7. Respiratory Depression and Hyperlocomotion

Male C57Bl/6J mice were tested for respiratory effects using automated, computer-controlled Comprehensive Lab Animal Monitoring System (CLAMS) apparatus. Mice (*n* = 8) were habituated in closed apparatus cages (23.5 × 11.5 × 13 cm^3^) for 30 min before being administrated with the hit i.p. at the experimented high dose of 3.0 mg/kg, and an even higher dose at 15 times of the ED_50_ value, 10.6 mg/kg, *i.p.* Mice were then returned to chambers, and their respiration rate (breaths/min) was measured for 100 min in 30 s intervals using a pressure transducer built into the enclosed, sealed CLAMS cage. The resultant change in pressure during mouse breathing was recorded to reveal the respiration rate (breaths/min).

Infrared beams located in the floor measured locomotion as the number of beam breaks. Respiration and locomotive data were averaged over 20 min periods for 120 min post-injection of the test compound. Data are presented as % vehicle response ± SEM, ambulation or breaths per minute.

### 4.8. Antinociceptive Tolerance and Naloxone-Precipitated Opioid Withdrawal Assay

Mice were randomly assigned to one of three groups and given saline (*i.p.; n* = 10), morphine (10 mg/kg, *i.p.*; *n* = 10) or CycloAnt (3 mg/kg, *i.p.*; *n* = 9) twice daily for 4 days, and once more on the morning of the fifth day. Mice were then tested in the 55 °C WWTW assay to assess for the loss of therapeutic efficacy (antinociceptive tolerance; see above). Two hours after this test, naloxone (10 mg/kg, *i.p.*) was administrated to all mice to precipitate opioid withdrawal [39]. Mouse activity was digitally recorded and evaluated later for an established record of withdrawal behaviors for 15 min [39]. Immediately upon injection of naloxone, all mice were scored for number of jumps, forepaw tremor, wet dog shakes, straightening, rearing, forepaw licking, teeth chattering, and the presence (as +/−) and number of soft stools.

### 4.9. Statistical Analysis

All dose–response lines were analyzed by regression, and ED_50_ (effective dose producing 50% antinociception) values and 95% C.I. were determined using individual data points from graded dose–response curves with Prism 10.0 software (GraphPad, La Jolla, CA, USA). Percent antinociception was used to determine within-group effects and to allow comparison to baseline latency in tail-withdrawal experiments. The statistical significance of differences between ED_50_ values was determined by evaluation of the ED_50_ value shift via nonlinear regression modeling with Prism 10.0 software. Significant differences in behavioral data were analyzed by ANOVA (one-way or two-way with repeated measures (RM), as appropriate). Significant results were further analyzed with Sidak’s, Tukey’s, or Dunnett’s multiple comparison post hoc tests, as appropriate. All data are presented as mean ± SEM, with significance set at *p* < 0.05.

## Data Availability

Data is contained within the article and the Appendix A.

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
