# Peer review of "Identification and Pharmacological Characterization of a Low-Liability Antinociceptive Bifunctional MOR/DOR Cyclic Peptide"

_molecules, 2023, doi:10.3390/molecules28227548_

Round 1
Reviewer 1 Report
Comments and Suggestions for Authors
Dear Authors,
The manuscript entitled Identification and Pharmacological Characterization of A Low-Liability Antinociceptive Bifunctional MOR/DOR Cyclic Peptide presents interesting data to be published by the journal with a grounded discussion. However, I suggest some specific changes for the authors.
In the introduction, the authors begin with the following sentence "Opioid analgesics are effective pain medications for relieving acute, severe pain". In fact, due to a large addiction crisis growing due to the consumption of opioids, I think that writing this generates a certain strangeness, as opioids are only used as a last resort for pain or for something very specific. I think the authors can make this very clear in their text before actually getting into opioids. They can start by defining pain, which medications can be used and then talk about what types of pain opioids can be used for.
I confess that I didn't understand the Figure 1, I would ask the authors to remake the Figure 1 and/or leave the structures that were most related to the work. Example of CycloAnt and CycloAnt-Leu. I could not see R1 in the figure 1
The authors in figure 7 do not show vehicle data. It is necessary to show the vehicle group to compare the effects
Table 1 has inconsistent information. Example: The presence of diarrhea must be a yes (+) or no (-); There are no 6.67 mice with diarrhea, I can imagine 6 animals with diarrhea but not 6.67. Other parameters such as Forepaw Tremor, Wet Dog Shakes, Straightening have broken values and the qualitative parameters are not presented in this way. You might use some scales for this.
I think that the authors' discussion in the part about equianalgesic doses of morphine has to be revised, due to the fact that the dose of one analgesic that is equivalent in pain-relieving effects to that of another analgesic; the dose in steady state providing the same analgesic response. In this part, the authors write and compare the adverse effects and not the antinociceptive effect (line 397)
Minor revision:
Line 89, ED 80? Is it okay this?
Line 308, the abbreviation of OIH should be described before
Why the authors used differents temperatures to evaluate hyperalgesia and tolarence?
Line 538 you should adjust the reference
Line 552-553, Did you used 8-16 mices? So big not?!
Line 586, The authors checked the number of soft stools or diarrhea, because they are different things
Comments on the Quality of English LanguageThe quality of English is good.
Author Response
I confess that I didn't understand the Figure 1, I would ask the authors to remake the Figure 1 and/or leave the structures that were most related to the work. Example of CycloAnt and CycloAnt-Leu. I could not see R1 in the figure 1.
—We modified Figure 1 by adding R1 = Gly and R3 = Dap(Ant) for clarification.
The authors in figure 7 do not show vehicle data. It is necessary to show the vehicle group to compare the effects
—Apologies for this omission. We have now added the vehicle (saline) data in Figure 7 for the requested comparison.
Table 1 has inconsistent information. Example: The presence of diarrhea must be a yes (+) or no (-); There are no 6.67 mice with diarrhea, I can imagine 6 animals with diarrhea but not 6.67. Other parameters such as Forepaw Tremor, Wet Dog Shakes, Straightening have broken values and the qualitative parameters are not presented in this way. You might use some scales for this.
— While qualitative scoring of symptoms is widely used to assess opioid withdrawal, these have faced criticism for possible observer bias. To forestall this concern in this study, we focused on quantifiable measures as utilized previously (see reference 39). Moreover, to increase rigor, we performed the testing in groups of 9 (for CycloAnt) or 10 (for saline or morphine) mice. The collected values in Table 1 thus represent the averaged responses (± SEM) across categories, facilitating statistical analysis, but resulting in “broken values” as the reviewer correctly notes.
The data in Table 1 is now edited to clarify these matters. Subject number is included in both the table and methods, and a description in the header clarifies reported values to be average and SEM. Specific to the reviewer’s concern, the former “diarrhea” category has been edited to the more-accurate “Number of soft stools.” Moreover, observation of diarrhea (as a yes or no value as requested by the reviewer) is now included.
I think that the authors' discussion in the part about equianalgesic doses of morphine has to be revised, due to the fact that the dose of one analgesic that is equivalent in pain-relieving effects to that of another analgesic; the dose in steady state providing the same analgesic response. In this part, the authors write and compare the adverse effects and not the antinociceptive effect (line 397)
— We concur that this passage does not compare analgesic efficacy. Our intent was instead to highlight the significant difference in adverse effects between the two opioids at a dose where the antinociception was equivalent. However, in response to the reviewer’s request, we have now edited this passage to clarify the point, but leave out the “equianalgesic” description.
Minor revision:
Line 89, ED 80? Is it okay this?
—Yes; this is as reported.
Line 308, the abbreviation of OIH should be described before
—Now described after Opioid-Induced Hyperalgesia in the line of the subtitle 2.9
Why the authors used differents temperatures to evaluate hyperalgesia and tolarence?
—These are well validated temperatures in tail-withdrawal assays for the respective effects measured. 55oC is a standard temperature for evaluating opioid-induced antinociception, and through it’s absence, antinociceptive tolerance. In contrast, hyperalgesia is best measured by a reduction in baseline tail withdrawal latency, and so, is best measured with slightly lower water temperatures such as 48oC. This is well-validated in the literature, and now reviewed in section 4.5 of the methods (citing references 50 and 51 to further justify these conditions).
Line 538 you should adjust the reference
—The reference has been adjusted (with thanks for catching this).
Line 552-553, Did you used 8-16 mices? So big not?!
—We used 8 mice to test each sample in the screening of the library. This is now corrected in the text (with apologies for the error).
Line 586, The authors checked the number of soft stools or diarrhea, because they are different things
—Updated; see our response to comments about Table 1 (above).
Reviewer 2 Report
Comments and Suggestions for Authors
The authors synthesized a mixture-based cyclic peptide library and screened it directly in vivo in the warm-water tail-withdrawal assay to identify compounds showing antinociceptive activity with intraperitoneal (i.p.) administration. As the authors rightfully point out, this direct in vivo screening permits the identification of compounds with some pharmacokinetic properties of interest for eventual drug development. The identified lead compound, CycloAnt, produced 6-fold more potent antinociception than morphine, did not produce respiratory depression and opioid-induced hyperalgesia, and induced less physical dependence than morphine. Thus, CycloAnt represents an interesting new lead compound for the further development of opioid analgesics with reduced side effects. The paper will be acceptable for publication after some minor revision, as outlined below.
1) The type of cyclic opioid peptide with an exocyclic Tyr residue and an adjacent D-Lys residue linked to the C-terminal carboxylate of the pentapeptide via its epsilon-amino group is based on previously published cyclic opioid peptides, such as Tyr-[D-Lys-Gly-Phe-Leu] (J. DiMaio et al., J. Med. Chem. 25, 1432-1438 (1982)), which should be cited in the paper. It is not surprising that the compound CycloAnt-Leu with an exocyclic Leu residue has much lower potency, since an exocyclic Tyr residue is required for good activity of this type of cyclic opioid peptide (similar to the requirement of an N-terminal Tyr residue in most endogenous opioid peptides)).
2) Page 12, top: It is mentioned that “pretreatment with a DOR-selective antisense oligonucleotide decreased morphine dependence (ref. 45)”. This was first demonstrated by Inturrisi’s group (B. Kest et al., Brain Res. Bull. 39, 185-188 (1996)) and this reference should also be cited. Also of importance in this context is the observation that morphine retained its MOR-mediated analgesic activity in DOR knock out mice without producing tolerance and dependence upon chronic administration (Y. Zhu et al., Neuron 24, 243-252 (1999)); this should be mentioned and referenced.
3) The authors may want to explain why CycloAnt increased breathing compared to the vehicle.
4) Page 12, line 406; The authors refer to CycloAnt as a MOR/DOR agonist here. However, as they point out in the Discussion, it acts as a DOR antagonist at a physiologically normal level of DOR expression. Shouldn’t it then be called a MOR agonist/DOR antagonist?
5) In the “Materials and Methods” section the authors should make a statement about the degree of purity of the synthesized peptides. Also, mass spec data for CycloAnt and CycloAnt-Leu are missing.
Comments on the Quality of English Language6) There are a number of minor grammatical deficiencies throughout the manuscript that need to be corrected.
Author Response
1) The type of cyclic opioid peptide with an exocyclic Tyr residue and an adjacent D-Lys residue linked to the C-terminal carboxylate of the pentapeptide via its epsilon-amino group is based on previously published cyclic opioid peptides, such as Tyr-[D-Lys-Gly-Phe-Leu] (J. DiMaio et al., J. Med. Chem. 25, 1432-1438 (1982)), which should be cited in the paper. It is not surprising that the compound CycloAnt-Leu with an exocyclic Leu residue has much lower potency, since an exocyclic Tyr residue is required for good activity of this type of cyclic opioid peptide (similar to the requirement of an N-terminal Tyr residue in most endogenous opioid peptides)).
—Added the reference as Ref.5. The numerical number of the references after Ref.5 were changed accordingly.
2) Page 12, top: It is mentioned that “pretreatment with a DOR-selective antisense oligonucleotide decreased morphine dependence (ref. 45)”. This was first demonstrated by Inturrisi’s group (B. Kest et al., Brain Res. Bull. 39, 185-188 (1996)) and this reference should also be cited. Also of importance in this context is the observation that morphine retained its MOR-mediated analgesic activity in DOR knock out mice without producing tolerance and dependence upon chronic administration (Y. Zhu et al., Neuron 24, 243-252 (1999)); this should be mentioned and referenced.
—We agree these are important precursors to the present findings, and have included them in the discussion. B. Kest et al., Brain Res. Bull. 39, 185-188 (1996)) was added as Ref. 48. We also included a sentence and cited the paper (Y. Zhu et al., Neuron 24, 243-252 (1999)) as ref.49. It is at line 412-414 and reads
‘It is also consistent with the study in DOR knock out mice that morphine retained its MOR-mediated analgesic activity without producing tolerance and dependence upon chronic administration.[49]’
3) The authors may want to explain why CycloAnt increased breathing compared to the vehicle.
—We agree this is an interesting observation, adding to a pattern also noted with the bifunctional MOR agonist/DOR antagonist [Nal(2’)4]CJ-15,208 (ref. 46). While a physiological study to better understand how DOR antagonism may increase the breathing rate is of interest, it is beyond the scope of this initial study to identify CycloAnt from the mixture-based library and characterize it. However, it is our hope that CycloAnt may be a useful tool in planned future studies to examine this phenomena.
4) Page 12, line 406; The authors refer to CycloAnt as a MOR/DOR agonist here. However, as they point out in the Discussion, it acts as a DOR antagonist at a physiologically normal level of DOR expression. Shouldn’t it then be called a MOR agonist/DOR antagonist?
—We apologize for the error (a result of editing earlier drafts). We have now correctly this to MOR agonist/DOR antagonist
5) In the “Materials and Methods” section the authors should make a statement about the degree of purity of the synthesized peptides. Also, mass spec data for CycloAnt and CycloAnt-Leu are missing.
—Purity and MS data added
Comments on the Quality of English Language
6) There are a number of minor grammatical deficiencies throughout the manuscript that need to be corrected.
—A careful edit of the manuscript has now been completed, with our apologies for any inadvertent grammatical errors that might remain. (Dr. McLaughlin notes he would not wish to disappoint Mr. Sullivan, his inspiring 8th grade English teacher and grammar expert extraordinaire.)
Reviewer 3 Report
Comments and Suggestions for Authors
This manuscript is about the identification of cyclic opioid peptide ligands that interact with MOR and DOR as an agonist and an antagonist, respectively, from a peptide library. The authors validated the identified ligand's efficiency in reducing respiratory effects and locomotive effects in vivo tests, indicating the therapeutic benefits of dual opioid activity as a safer analgesic.
Overall, this manuscript reads well, and the reviewer recommends this manuscript be published in Molecules. Here are the minor revisions the reviewer suggests.
- The positional screening was done in 91 samples composed of linear pentapeptides. Nonetheless conserving important residues, Dap(Ant) at position 3 and Gly at position 5, the peptides are linear and quite different in structure from cyclic ones. The authors should state the inconsistency caused by the rough screen.
- Typo in the Abbreviations: "Comprehensive Lab AnimalMonitoring" should be revised to "Comprehensive Lab Animal Monitoring System"
- In Abbreviations: KO stands for knockout but not for gene-distrupted mice.
Author Response
Comments and Suggestions for Authors
This manuscript is about the identification of cyclic opioid peptide ligands that interact with MOR and DOR as an agonist and an antagonist, respectively, from a peptide library. The authors validated the identified ligand's efficiency in reducing respiratory effects and locomotive effects in vivo tests, indicating the therapeutic benefits of dual opioid activity as a safer analgesic.
Overall, this manuscript reads well, and the reviewer recommends this manuscript be published in Molecules. Here are the minor revisions the reviewer suggests.
1. The positional screening was done in 91 samples composed of linear pentapeptides. Nonetheless conserving important residues, Dap(Ant) at position 3 and Gly at position 5, the peptides are linear and quite different in structure from cyclic ones. The authors should state the inconsistency caused by the rough screen.
—We agree with the reviewer on that the complexity of the products in the library should be stated. We added a paragraph at lines 368-379 to the discussion. It reads
‘The mixture-based cyclic peptide library is constituted using imidazole-catalyzed cyclization of linear pentapeptide. While the goal is to cyclize all the linear peptides in a head-to-tail pattern, however, the composition of the mixture-based positional library is more complicated as side-to-tail cyclization can take place when Lys or D-Lys is in the linear sequence. Lys and D-Lys are used at positions 2 and 4, and Lys is used at position 5 in the linear samples; therefore, the peptides in the library are not only cyclized in the head-to-tail pattern, but also has the side-to-tail as a side product when Lys or D-Lys is in the sequence. Since we are tracking the antinociceptive effect not the purity of the compound in the library, as long as the side reaction is reproducible, the active compounds can be identified through the deconvolution of the mixture library. In this case, CycloAnt, a side-to-tail cyclic peptide, which produced potent antinociceptive effect has been successfully identified through the phenotypic screening in mice.’
2. Typo in the Abbreviations: "Comprehensive Lab AnimalMonitoring" should be revised to "Comprehensive Lab Animal Monitoring System"
—Corrected
3. In Abbreviations: KO stands for knockout but not for gene-distrupted mice.
—Corrected to gene-knockout mice
Round 2
Reviewer 1 Report
Comments and Suggestions for Authors
Dear authors,
I read what you wrote and I ask you to rewrite it here "Opioid analgesics have long been the gold standard for effective relief of acute, severe pain as well as chronic pain"
I believe that clinically, opioids are used for the relief of moderate to extreme pain. Strong opioids are often prescribed for more severe pain – such as after surgery and accidents – or to treat pain in people with cancer. Acute pain is not recommended using opioids
Author Response
Open Review
( ) I would not like to sign my review report
(x) I would like to sign my review report
Quality of English Language
( ) I am not qualified to assess the quality of English in this paper
( ) English very difficult to understand/incomprehensible
( ) Extensive editing of English language required
( ) Moderate editing of English language required
( ) Minor editing of English language required
(x) English language fine. No issues detected
|
Yes |
Can be improved |
Must be improved |
Not applicable |
|
|
Does the introduction provide sufficient background and include all relevant references? |
( ) |
(x) |
( ) |
( ) |
|
Are all the cited references relevant to the research? |
(x) |
( ) |
( ) |
( ) |
|
Is the research design appropriate? |
(x) |
( ) |
( ) |
( ) |
|
Are the methods adequately described? |
(x) |
( ) |
( ) |
( ) |
|
Are the results clearly presented? |
(x) |
( ) |
( ) |
( ) |
|
Are the conclusions supported by the results? |
(x) |
( ) |
( ) |
( ) |
Comments and Suggestions for Authors
Dear authors,
I read what you wrote and I ask you to rewrite it here "Opioid analgesics have long been the gold standard for effective relief of acute, severe pain as well as chronic pain"
I believe that clinically, opioids are used for the relief of moderate to extreme pain. Strong opioids are often prescribed for more severe pain – such as after surgery and accidents – or to treat pain in people with cancer. Acute pain is not recommended using opioids
- Following the content added earlier in response to their previous demand for text introducing the nature of pain and transmission of nociception, we now edit lines 38-41 to accommodate the reviewer’s current request as follows:
“Opioid analgesics have a long history of use to relieve moderate to extreme pain [1]. However, repeated opioid use results in clinical liabilities such as the development of tolerance and dependence and a potentially lethal respiratory depression [2].”
Submission Date 13 October 2023
Date of this review 03 Nov 2023 21:14:08